# Patient Preferences in Breast Cancer: A Scoping Review

**DOI:** 10.3390/cancers18010134

**Published:** 2025-12-31

**Authors:** Charlotte Verbeke, Fiene Schuermans, Fien Vanopré, Aline Belmans, Maxime Van Houdt, Patrick Neven, Isabelle Huys

**Affiliations:** 1Department of Pharmaceutical and Pharmacological Sciences, KU Leuven, 3000 Leuven, Belgium; 2Division of Gynaecological Oncology, Department of Obstetrics and Gynaecology, University Hospitals Leuven, KU Leuven, 3000 Leuven, Belgium

**Keywords:** breast cancer, patient preferences, patients’ needs, patient involvement, recruitment strategies, healthcare decision-making, scoping review

## Abstract

Patients with breast cancer are all confronted with different needs and preferences concerning their treatment and care. Understanding what matters to breast cancer patients can help tailor decisions. This scoping review aims to provide an overview of published studies identifying breast cancer patient preferences. The review shows how the diversity within and between the preference studies complicates a robust and standardized implementation of preferences in decision-making processes. The article suggests how stakeholders can create valuable impact for and with breast cancer patients, for example, by linking preference studies to clear objectives and implementation strategies, and by involving patients throughout the whole process.

## 1. Introduction

Breast cancer is the most common cancer among women worldwide and remains a leading cause of mortality in this population [1]. Both the disease and its treatments profoundly affect patients’ overall survival as well as their quality of life due to a wide range of symptoms and side effects, necessitating a multidisciplinary approach to care. To fully understand the needs and experiences of patients, information on patient experiences, alongside clinical trial evidence on physical parameters, is necessary.

One type of patient experience data is data on patient preferences, demonstrating what matters most to patients in terms of their experiences with treatments or disease characteristics, and which trade-offs (e.g., between treatment benefits and risks) patients are willing to make. Such data result from ‘patient preference studies’ (PPS) [2].

Methods used in patient preference studies can be categorized as qualitative (preference exploration) and quantitative (preference elicitation) [2]. Qualitative methods aim to explore patients’ perspectives in depth; examples are interviews and focus group discussions. Quantitative methods aim to collect quantifiable data and can be grouped into four categories: (i) discrete choice-based methods, (ii) ranking methods, (iii) indifference techniques, and (iv) rating methods [2,3]. In the paper of Soekhai et al. [3], an overview of the different methods in these four categories can be found.

The different methods offer distinct advantages for capturing the complexities of patient preferences and to guide and support patient decision-making.

Besides the importance of choosing the adequate method to answer the specific research questions of the patient preference study, other important elements (such as the heterogeneity in the responses of patients on preferences in view of their demographic backgrounds, or particular recruitment strategies to include patients in the study) should be considered.

To study patient heterogeneity, different techniques can be used, e.g., sub-group analyses, mixed logit models, and latent class analysis [2,4,5]. To determine preferences from heterogenous samples of breast cancer, patients’ effective recruitment strategies are crucial. In the literature review of Wang et al. [6], effective recruitment strategies for adolescent and young adult (AYA) cancer survivors were identified, showing that for this target group, internet and social networking were the strategies mainly used, and with a higher participation rate compared to other more conventional methods (such as recruitment via hospitals).

This scoping review aims to provide a comprehensive overview of the design and outcomes of patient preference studies conducted among breast cancer patients concerning their disease, treatment, and care, and to (i) provide insights on the elements assessed in the preference studies, (ii) investigate if variations in patient preferences (preference heterogeneity) are considered, (iii) analyze recruitment strategies, and (iv) identify patient involvement and communication strategies.

## 2. Methods

The methodology of the scoping review was predefined and documented in a registered protocol at the Open Science Framework (OSF) database (registration DOI: 10.17605/OSF.IO/M2H4K). The review was compliant with the PRISMA-ScR guidelines for scoping reviews (Appendix A).

### 2.1. Search Strategy

Five databases were systematically searched: PubMed, Embase, Web of Science, CINAHL, and Scopus. The search string was developed by the research team and verified and further refined with a biomedical information specialist. The search strategy was based on two key concepts: ‘Breast cancer’ and ‘Patient preference’. Relevant synonyms and related terms were included, combined with Boolean operators to ensure an optimal search strategy. The detailed search strategies can be found in Appendix A. The search was conducted on 9 October 2024. The articles were extracted from all databases, added and manually deduplicated in the Rayyan Software [7].

### 2.2. Study Selection

#### 2.2.1. Inclusion and Exclusion Criteria

Inclusion criteria were (i) studies involving breast cancer patients or breast cancer survivors (population), (ii) qualitative and/or quantitative patient preference studies (study design) eliciting patient preferences concerning disease, treatments, and care (concept), and (iii) publication in English. Exclusion criteria were (i) studies not involving breast cancer patients or involving a broader population, e.g., a mixed population of breast cancer patients and healthcare professionals, (ii) all other study designs than patient preference studies (e.g., reviews, retrospective data analysis, and clinical trials), (iii) studies with another aim than the elicitation of patient preferences, (iv) papers where the full text was not available, preprints, conference abstracts, conference proceedings, and book chapters, and (v) publications that were not in English. A detailed overview of the inclusion and exclusion criteria can be found in Appendix A.

#### 2.2.2. Screening Process

The screening of the articles was conducted double-blind in two phases. Prior to the screening phase, a pilot step assessing a limited number of articles on eligibility was performed by independent researchers (AB, FS, and FV; double-blind with CV) to ensure alignment and refine the inclusion and exclusion criteria where needed. In the first screening phase, researchers AB, FS, and FV each reviewed one third of the articles, while researcher CV screened all the articles on title and abstract. Disagreements were discussed and resolved among the research team (AB, FS, FV, and CV). Then, in the second screening phase on the full text, researchers FS and FV each assessed half of the articles, and researcher CV assessed all articles. Again, all disagreements were resolved by consensus among the researchers (FS, FV and CV), and the selection of the studies for data extraction was finalized.

### 2.3. Data Extraction and Analysis

Data extraction was performed using a predefined extraction table template. A pilot phase was performed prior to the actual data extraction, to ensure that all relevant elements were extracted and interpreted in the same way among the research team. During the subsequent consensus meeting, the extraction table template was refined and finalized (Appendix A). The data extracted was related to (i) general characteristics of the publication, (ii) characteristics of the patient preferences study, (iii) patient heterogeneity, and (iv) patient involvement, recruitment, and communication strategies. Researchers FS and FV extracted the data from half of the articles; researcher CV extracted the data from all of the articles independently to ensure consistency. The extracted raw data was consolidated in an Excel spreadsheet, categorized based on predefined themes, and summarized for further analysis. Descriptive analysis of the extracted data was performed by FS, FV, and CV.

## 3. Results

### 3.1. Study Inclusion

A total of 19.017 articles were identified through the database search, from which 10.511 duplicates were removed. A total of 8.506 unique articles were screened on title and abstract in the first phase. A total of 8.261 articles were excluded based on the eligibility criteria. Eight reports were not retrieved, resulting in 237 articles eligible for full-text screening in a second screening phase. Thirty-one articles met the inclusion criteria and were analyzed in the review, as shown in the flow diagram (Figure 1).

### 3.2. Study Characteristics

The study characteristics of the 31 studies are summarized in (Appendix A), including general characteristics of the publication (year, first author, country of the first author, journal, and funding) and general characteristics of the patient preference study (objectives, PPS methods, ranking/grading elements). For most of the studies, the country of the first author was the USA (*n* = 11). The year of publication ranged from 1995 to 2024.

#### Patient Characteristics

As stated in the inclusion criteria, both studies involving breast cancer survivors and/or breast cancer patients were included. In Table 1, the types of breast cancer of the participants in the studies are summarized. In some studies, patients across different breast cancer stages and with different subtypes were included.

An overview of the study population characteristics can be found in Appendix A, including the inclusion and exclusion criteria, breast cancer types of the study population, and information on the age and country/ethnicity of the included participants.

### 3.3. Patient Preference Studies

#### 3.3.1. Steps Prior to the Patient Preference Studies

Prior to the actual assessment of what patients prefer or value most in view of their treatment or disease characteristics, most of the studies (*n* = 21) (Table 2) included literature reviews to identify studies on breast cancer disease or treatment characteristics to be used in the preference study as starting points [8,10,11,13,14,15,16,17,18,19,20,22,23,24,25,27,28,29,36,37,38]. Two studies mentioned reviewing breast cancer web forums to verify if the selected side effects were influencing patients’ daily life [11,16].

Moreover, in seven studies, clinicians or experts in the field were involved in the selection or validation steps [8,10,16,17,19,25,27]. In two studies, an advisory board, consisting of clinicians, breast cancer patients, and former health policy makers, finalized the characteristics included in the preference studies [13,14].

Also, in some studies (*n* = 7), interviews or focus group discussions with patients and/or clinical experts were conducted, to provide feedback, to verify, and to refine elements of the study designs [15,18,20,22,24,28,33]. The study of Chou et al. [19] mentioned that non-medical people reviewed the descriptions for each health state used in the study.

Twelve studies mentioned conducting a pilot study [11,13,14,15,21,23,24,26,27,33,37,38]; in nine cases, patients were involved in this step [11,14,15,23,24,26,33,37,38]. In two studies, the pretest of the questionnaire was performed with the general population [13,27], to ensure that patients hard to recruit would participate in the actual study instead of the pilot phase [13].

#### 3.3.2. Patient Preference Methods

Patient preferences were obtained through various methods (Table 2). In the study of Silva et al. [17], preferences were explored via interviews and ranked by importance in an online survey. In three studies, hypothetical scenarios of treatments were presented to patients and discussed during an interview [30,35,37].

The method used in the other studies could be classified in one of the elicitation categories as outlined in the PREFER recommendations [2]. The study methods used are distributed as followed: (i) twelve studies using discrete choice-based methods (discrete choice experiment [13,14,15,18,20,23,26,27,38] and adaptive conjoint analysis [16,24,28]); (ii) six studies using ranking methods (choice-based conjoint [12,21,22,25,29] and Q-sort task [36]); (iii) eight studies using indifference methods (standard gamble [10,11], threshold task [8], time trade-off [9,19], and trade-off technique [31,32,33]); and (iv) three studies using rating methods (visual analog scale [10,19] and analytical hierarchy process [34]).

#### 3.3.3. Ranking or Grading Elements

In all the studies, patients had to rank or grade certain (treatment) elements and/or hypothetical (treatment) scenarios. An overview of the ranking/grading elements can be found in Table 2.

Such elements presented in these studies are based on characteristics of breast cancer therapies (for instance, nausea and fatigue due to chemotherapy [11,16], or joint and muscle pain and libido decrease due to endocrine therapies [28]). In five studies, the elements were related to endocrine therapies [10,28,32,35,36]; one study presented elements related to radiotherapy [31], and another one to targeted therapy [23]. In seven studies, the elements were based on chemotherapy [8,9,11,16,21,29,33].

In the study of Omori et al. [26], the attributes were selected based on findings from the MONARCH 2 trial [39], with elements related to the treatment categories targeted agents and endocrine therapy. Also, in the study of Nazari et al. [27], elements were identified starting from endocrine and targeted therapies. Different hypothetical scenarios, consisting of chemotherapy and endocrine therapy, were used in the study of McQuellon et al. [30].

The elements in three of the preference studies were not formulated based on treatments. In one study, characteristics of an axillary dissection were used as starting point for the elements in the preference study [37]. In another study, the preferences of centrale venous devices are examined [38]. The third study based the elements on the treatment aspects currently considered in value frameworks [20].

The elements evaluated in the studies can be classified as (i) adverse events(related to the treatments) (*n* = 23) [8,10,11,12,13,14,15,16,17,18,19,20,21,22,23,24,25,26,27,28,29,30,34], (ii) effectiveness/efficacy (*n* = 4) [21,22,28,34], (iii) life expectancies and survival (*n* = 14) [8,13,14,17,18,20,23,24,26,27,32,33,35,37], (iv) quality of life (*n* = 7) [12,13,14,17,22,24,38], (v) administration (*n* = 7) [8,16,21,22,26,27,28], (vi) breast-cancer-related disease stages (*n* = 5) [10,19,29,31,37], (vii) costs (*n* = 8) [12,13,14,18,20,25,27,38], and (viii) other (*n* = 6) [8,12,15,20,37,38] (Appendix A).

Adverse events that appeared in several studies were as follows: diarrhea (*n* = 10) [11,15,16,19,22,23,24,25,26,34], nausea/vomiting (*n* = 9) [11,15,16,19,22,23,24,25,34], fatigue (*n* = 8) [11,15,16,19,22,24,25,34], and alopecia (*n* = 6) [11,16,22,24,25,34]. The study of Omori et al. [26] included three attributes in the DCE on diarrhea: (i) frequency of stools, (ii) incidence of diarrhea, and (iii) duration of diarrhea.

The attribute that appeared most in the category ‘life expectancies and survival’ was progression-free survival (*n* = 6) [14,17,18,23,26,27]. Seven studies included elements related to ‘quality of life’ [12,13,14,17,22,24,38].

### 3.4. Preference Outcomes

An overview of (i) the (statistical) methods to obtain preferences and (ii) some of the preference results can be found in Appendix A. If elements of life expectancies and survival were included, these elements were in most studies evaluated as important [8,14,17,20,24,26,27,32,33,35,37]. In the studies with elements on quality of life, it seemed that in some studies, these elements influence patients’ decisions and were ranked as important [12,13,14,17,24]. The preference outcomes are very dependent on the elements that were evaluated in the study and the design of the study.

In 24 of the included studies [8,9,10,11,12,13,14,15,19,20,21,23,24,26,27,28,29,30,31,32,33,35,37,38], preference heterogeneity was assessed. Statistical analyses were performed to assess heterogeneity, and the heterogeneity results are also summarized in Appendix A.

### 3.5. Patient Recruitment and Patient Involvement

The studies included in this review aim to identify which elements in their treatment or disease matter (most) to patients, and therefore recruiting patients willing to participate is needed. Different recruitment strategies were described. Moreover, the involvement of patients is crucial in patient preference studies. Table 3 provides an overview of patient recruitment strategies and patient involvement in the selected studies. Patient involvement is divided into two parts: (1) patients as participants, including communication to patients, and (2) patients as co-researchers and their involvement throughout the study (from the study design and pilot studies until the interpretation of the results).

#### 3.5.1. Patient Recruitment

##### Recruitment Strategies

The recruitment strategy used most was via hospitals and cancer centers (*n* = 13) [9,10,11,15,16,18,19,27,28,32,33,36,38]. One of the studies also mentioned community pharmacies [28].

In some of the studies, eligible participants were recruited via personal contact by a researcher or physician (*n* = 5) [15,24,30,34,37]. In two studies, information was sent to patients, followed by a phone call if patients were open to participate [35,37]. Also, in the study of McQuellon et al. [30], eligible patients were called by the study coordinator, for the explanation and invitation to participate.

Another recruitment strategy frequently mentioned was through online channels (*n* = 10) [12,15,17,21,22,23,24,29,34,36]. One of the studies used internet advertisements [34].

In some studies, the (online) distribution of the survey was facilitated by a third party; (i) patient support organizations (*n* = 7) [12,17,21,24,28,29,36] and (ii) (market research) companies (*n* = 5) [13,14,20,23,29].

In the study of DaCosta DiBonaventura et al. [22], invitations were sent to panelists of cancer-specific online panels. In two studies, invitations were sent via email after identification from a panel [25,26].

In three studies, a combination of different recruitment channels was applied [15,24,34].

Some studies (*n* = 6) also described the identification of eligible patients: via the hospital pharmacy [10], via a database [35,37], by the treating physician [30], or via a (web) panel [25,26]. In two of the studies, the recruitment strategy was not specified [8,31].

#### 3.5.2. Patient Involvement

##### Involvement in Study Design

In 14 of the studies [11,13,14,15,18,20,22,23,24,26,28,33,37,38], certain roles of patients as co-researchers were identified. There are two categories in how patients were involved in the studies:

(a)Patient providing advice and input

In two studies, patients are part of an advisory board in the study [13,14]. In one study, patient advocacy groups provided input on the study protocol [23]. Moreover, qualitative interviews and focus groups were conducted in five studies to provide insights and verify attributes [18,20,22,24,28].

(b)Pretest/pilot test/think-aloud interviews

In nine studies, the study was pretested with patients [11,14,15,23,24,26,33,37,38].

### 3.6. Evaluation and Future Preference Research

The strengths, limitations, and opportunities for future research described in the papers are summarized in (Appendix A). Some of the elements are explained more in detail below.

Different limitations on preference research in breast cancer could be identified. The first category of limitations is related to the sample in the preference studies. Several studies reported that the sample in their preference studies was small [8,10,15,17,18,20,27], homogeneous [20,37], and/or not representative [9,12,17,21,28,32,37,38]. Some studies highlighted that due to these sample limitations, it was not possible to generalize the results [20,23,25,26,33,35].

Moreover, selection biases [8,16,36] were mentioned due to the choices in, for example, recruitment, as part of convenience sampling [10,19,22]. In one of the studies [10], only outpatients were recruited, which may create a selection bias towards patients with better health and mobility.

Studies recruiting via an online setting [23] mentioned (i) bias towards patients that have access to technology, are technology savvy, and have better health [11,14,26], and (ii) that it may lead to illogical responses [11].

A second category of limitations relates to the design of preference studies. In the study of Tan et al. [10], a starting point bias was mentioned, because of the design where 50% perfect health was given as starting point, creating a narrower range of preferences.

Concerning the choice of attributes and the descriptions in the preference studies, some considerations were highlighted: (i) the higher the number of attributes, the greater the burden on respondents [20]; (ii) in the case of a limited number of attributes, preferences might be different if other attributes would be included [12,15,18,22,23,25,26,38]; (iii) some attributes may be difficult to understand [15,38]; (iv) descriptions might not be representative for a real-life situation [11,16]; and (v) attributes could not be deselected [24].

Moreover, the limitations related to hypothetical scenarios often make it impossible for patients to know which choices would be made in actual therapeutic decisions [18,20,23,29].

In some studies, patients themselves reported the data [12,20,22,23], which can lead to recall biases and self-presentation effects [22].

Only few studies (*n* = 9) clearly highlighted strengths of their study, both on the design [8,9,10,14,15,26,28,34,36] and on the sample [26,36].

## 4. Discussion

### 4.1. Points to Highlight from the Included Breast Cancer Preference Studies

#### 4.1.1. Various Preference Designs and Outcomes

The findings of this scoping review highlight the wide variety in which patient preference studies among breast cancer patients are conducted and the preference results are obtained. The 31 included studies are published during a wide time range (1995–2024), across a diverse range of countries.

The methods mostly used in the included studies (*n* = 12) were discrete choice-based methods (DCEs and adaptive conjoint analysis). As highlighted in previous literature reviews, DCEs are widely used in health economics, and this number continues to grow [40,41,42]. This is in line with the increasing recognition of the importance of obtaining and integrating patient preferences in decision-making as seen by various stakeholders [43,44].

Various ranking and grading elements were identified across the selected studies; most of them were based on adverse events of treatments (*n* = 23) and survival elements (*n* = 14). The descriptions of the survival elements were distinct. In one of the studies, it was mentioned that including both progression-free survival and overall survival as elements in a preference study was confusing for patients. In some studies, the effectiveness (*n* = 4) of the treatments was added as attribute to evaluate the overall survival. If a survival element was added, it was valued in almost all cases as (the most) important. Quality of life elements were only included in seven of the studies. As has been mentioned in the scoping review on patient preference in metastatic breast cancer by Bland et al. [45], often the survival elements and superior treatment benefits are concluded to be the most important to patients. Quality of life elements are not as often included in the studies, so it is not possible to assess the value.

#### 4.1.2. Selection and Formulation of Attributes

Besides the selection of attributes, the importance of including clearly formulated and understandable attributes for patients in a preference study is crucial. As reported as one of the limitations by Liu et al. [38], there might be difficulties in understanding the chosen attributes. In the study of Chou et al. [19], the health states were drafted without input from patients.

The qualitative (preference exploration) phase in patient preference studies can serve as a starting point to determine which elements are important for patients, and based on these insights, attributes can be formulated. However, as highlighted in the study of Clark et al. [40], conducting patient preferences studies is on the rise, but a decline is observed in the qualitative phases of a preference study that is aimed to inform attribute selection.

Moreover, the study of Reinisch et al. [24] mentioned, as one of the limitations, the fact that attributes could not be deselected. This again shows the relevance and importance of defining meaningful attributes, and highlights the crucial previous steps of a patient preference study where literature is screened and patients are consulted via interviews or focus groups.

Also, advisory boards consisting of multiple stakeholders could help to create and formulate adequate attributes. In only two of the included studies, an advisory board was involved in the study.

#### 4.1.3. Patient Recruitment Strategies

Almost all of the studies provided some insights on their recruitment strategy. However, often it remained vague and somewhat unclear. The main recruitment channel was via hospitals and cancer centers. As has been highlighted by Tomiwa et al. [46], different digital tools can help to target a more representative sample and to enhance the diversity of the patients participating. An evolution towards digital channels was observed, as well as the inclusion of patient organizations and market research companies.

Yet, all these recruitment strategies innately bring with them the risk of selection bias. To overcome the biases and move towards a better representation of (breast cancer) patients, hybrid recruitment strategies are suggested.

#### 4.1.4. Opportunities for Future Research

The main opportunities mentioned in the studies relate to the way information could be used in decision-making. Moreover, the importance of patient data was mentioned several times; however, clear implementation strategies and actions for the inclusion of preference data in decision-making are still lacking.

### 4.2. Suggestions for Patient Preference Studies in Breast Cancer

Different guidelines and recommendations are currently available on the conduct of patient preference studies. The PREFER recommendations provide guidance on how, when, and why to assess and use patient preferences [2]. Also, the ISPOR task force provides a roadmap to increase the usefulness and the impact of patient preferences in decision-making [47].

Below, some suggestions for patient preference studies among breast cancer patients are outlined.

(1)Standardization of methods and designs making findings comparable

A variety of preference designs and outcomes were identified, making it hard to know what matters most and how findings can be compared among studies. Furthermore, all the studies were conducted within a certain context, year, country, and type of breast cancer.

Suggestions would be (i) making choices in methods and design with a clear strategy and aim of implementation in specific decision-making processes (decisions for clinical trial designs, regulatory decision-making, HTA decision-making, and decisions in clinical practice), and (ii) consulting upfront the decision-makers involved to enhance the value of the data and impact on decisions.

(2)Selection of the attributes to be included

Many of the studies added a survival element that was often highly valued by patients. This should not be a surprise, given the fact that many women with breast cancer want to fight and survive, and want to receive treatments that prolong their life expectancies as much as possible, especially in the earlier phase of breast cancer.

Suggestions would be: (i) focusing on elements beyond survival when conducting a patient preference study, (ii) looking at elements that patients might accept but that make them suffer during treatments and impact patients’ quality of life, and (iii) treating data on patients’ needs and patient perspectives as needed and crucial next to clinical trial data to optimize decisions (a suggestion for decision-makers).

(3)Representative sample: for the chosen target or for the potential population

Different breast cancer stages and subtypes were at hand in different studies. However, in specific studies, the sample was often not representative for the eligible population of the study, so it is not possible to establish generalizability of results.

Suggestions would be (i) clearly defining which sample is needed based on the research questions, (ii) creating a hybrid recruitment strategy (combining traditional and digital channels), (iii) focusing the recruitment strategy on how to include more diverse and underrepresented patient groups, and (iv) considering appropriate compensation that might also enhance the recruitment of underrepresented groups.

(4)Preferences from experiences vs. hypothetical preferences.

Preference designs are created with the idea to determine what patients would choose, and which tradeoffs patients would be willing to make in hypothetical scenarios. When targeting breast cancer patients, some patients might have already experienced the side effects mentioned, and others might not.

Suggestions would be (i) asking patients to indicate their experiences with the attributes in the study, and (ii) considering targeting a broader population. Given the hypothetical scenarios, it might be considered to target all potential breast cancer patients.

(5)The concept of preference heterogeneity

It seems that different options related to treatment elements exist among patients, which may be influenced by the demographic or treatment or be disease-related (early vs. metastatic) or other characteristics of patients (preference heterogeneity). Such preference heterogeneity is assessed via a diverse range of ways:(i)In subgroup analyses, subgroups are defined based on patients’ characteristics starting from the sample. Associations between preferences and patients’ characteristics are tested.(ii)In latent class analyses, patients are segmented based on attribute importance patterns. Patients’ characteristics correlated with the class assignment can be determined (i.e., hidden elements that determine preferences).

Suggestions would be (i) to define upfront which heterogeneity to determine, (ii) to see if the results (for example, hidden preferences) will be able to inform the decision-making processes, and (iii) discuss upfront with decision-makers involved how the data will be treated.

(6)Information and communication with the included patients

Attention should be given to the patients participating in the study. The information that was given to patients prior to participating in the preference study was often not clearly reported and remained vague. Moreover, in none of the studies were patients informed about the results after the study. In the interview study of van Overbeeke et al. [48], the value of involving patients in all steps was underscored.

Suggestions would be (i) emphasizing the objectives of the study to patients, (ii) providing information via explanatory videos and interactive briefings, and (iii) systematically informing patients of the study results to improve transparency between the patients and researchers.

(7)Patient involvement as co-researchers

Involving patients from the start of the preference studies improves the acceptability, comprehensibility, and relevance of the study.

Suggestions would be (i) setting up an advisory board with all the different stakeholders involved (including patients), (ii) asking for feedback and input from patients on protocols and study designs, (iii) focusing on the formulation of attributes and how patients interpret and answer the questions, and (iv) pilot testing the study with patients.

### 4.3. Strengths and Limitations of the Scoping Review

Both the strengths and limitations for this scoping review could be highlighted. The study offers several strengths concerning design. Firstly, the research team collaborated with Research Library 2Bergen Desiré Collen, who were providing support throughout the process, helping with the search strategies, and providing articles that could not be retrieved. Secondly, five databases were used in the scoping review, allowing us to obtain as many relevant articles as possible.

Moreover, throughout all steps of the research, a double-blind approach was applied and consensus meetings to resolve disagreements were organized, aiming to reduce the risks of including or excluding articles and misinterpretations of the data.

Finally, the research teams’ choice to exclusively focus on the patients, and hence only include patient preference studies that exclusively deal with breast cancer patients or breast cancer survivors, allowed for the exploration of patients’ own preferences, choices, and experiences, leading to more authentic and patient-centered insights. This can also be seen as a limitation, since the perspectives of other actors was not analyzed, such as healthcare professionals who obviously may play a significant role in treatment decision-making.

Some other limitations should be acknowledged. Firstly, not all articles were available in full text, which may have limited the depth and completeness of the literature review. Moreover, only English-language articles were included, which may introduce a language or cultural bias, as important studies in other languages have been excluded. Furthermore, even though a double-blind approach was performed over the whole process, other researchers might still have made other decisions concerning the inclusion or exclusion of the papers and the data extracted. Also, in terms of the interpretation and reporting of the data, the researchers made choices regarding what to report and how to structure and categorize the findings of the included studies.

## 5. Conclusions

This scoping review provides an overview of different preference studies among breast cancer patients. The findings show diversity, both in terms of the methods and designs used, as well as the preference outcomes. Moreover, differences in preferences among patients were observed. Given this diversity within and between the different studies, drawing general conclusions remains challenging. This could also potentially slow down the clear and standardized implementation of patient preference studies in decisions. The proposed suggestions for patient preference studies in breast cancer aim to provide ideas for studies targeting elements with an implementation plan in mind, to create impact with the valuable insights patients provide.

## Figures and Tables

**Figure 1 cancers-18-00134-f001:**
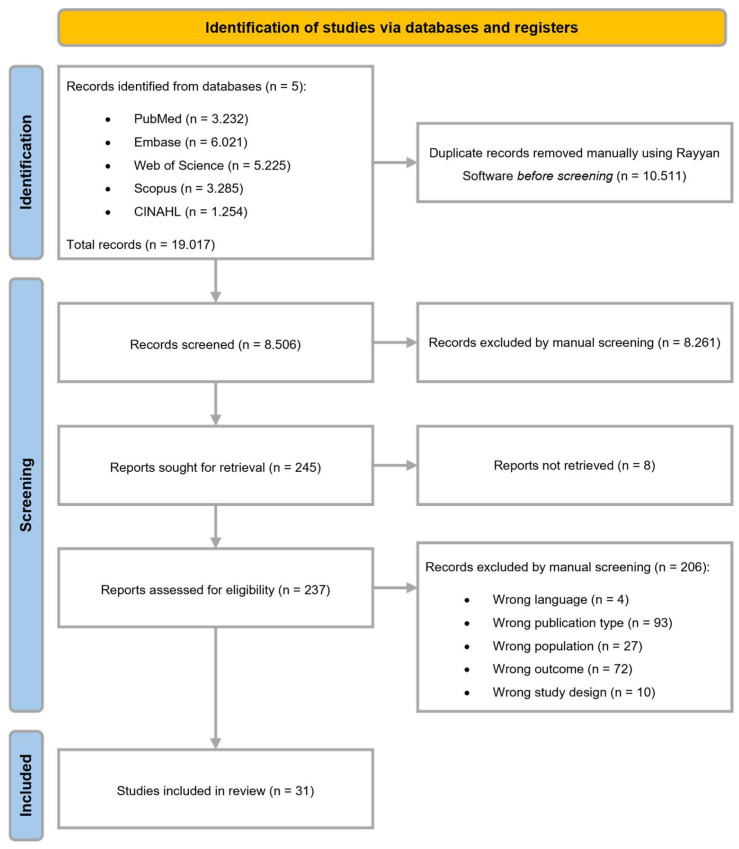
Flow diagram.

**Table 1 cancers-18-00134-t001:** Type of breast cancer of the participants in the study.

Type of Breast Cancer	Studies (*n*)	References
Breast cancer **survivors**	*n* = 1	[8]
**Different** disease **statuses** (disease free, local relapse, distant/relapse)	*n* = 1	[9]
**Varies** breast cancer **stages** (early and metastatic)	*n* = 9	[10,11,12,13,14,15,16,17,18]
**Different stages** of breast cancer **at diagnosis**	*n* = 1	[19]
Only **advanced or metastatic** breast cancer	*n* = 6	[20,21,22,23,24,25]
- **Locally advanced or metastatic HR+/HER2**−	*n* = 1	[24]
**HR+** patients in **different disease stages** **Estrogen receptor- positive** breast cancer	*n* = 2 *n* = 1	[26,27] [28]
**Early breast cancer** patients	*n* = 8	[29,30,31,32,33,34,35,36]
- Focus on **subtype HER2−**	*n* = 1	[29]
- **Premenopausal** early breast cancer patients	*n* = 1	[35]
**Early breast cancer** patients (stage I and II) and patients with a **diagnosis of ductal carcinoma in situ**	*n* = 1	[37]
Breast cancer stage or subtypes **not specified**	*n* = 1	[38]

**Table 2 cancers-18-00134-t002:** Methods and ranking/grading elements of the patient preference studies.

Study	Steps Prior to PPS	PPS Method	Type of Treatment Category the Elements are Based On	Ranking/Grading Elements
Tan et al., 2014 [10]	- Literature review - Validation by an oncology expert panel	Face-to-face interviews: assessing health states using the **visual analog scale** (VAS) and the **standard gamble method** (SG)	Endocrine therapy	- Assessment of 14 health states illustrating ○ common endocrine therapy-associated adverse effects ▪ Cataract ▪ Hip fracture ▪ Wrist fracture ▪ Spine fracture ▪ Vaginal bleeding ▪ Hot flushes ▪ Musculoskeletal disorder ▪ Pulmonary embolism ▪ Endometrial cancer ▪ Deep vein thrombosis ▪ Ischemic cardiovascular events ○ breast cancer-related disease stages ▪ No recurrence ▪ Locoregional recurrence ▪ Distant recurrence ▪ New contralateral breast cancer
Srikanthan et al., 2019 [8]	- Literature - Expert consensus	Face-to-face interviews: **threshold task** and post interview debriefing questionnaire	Chemotherapy	- Two hypothetical scenarios to assess strength of preference for chemotherapy ○ Treatment schedule ○ Follow up ○ Short term side effects ○ Possible long term side effects ○ Survival benefit ○ Fertility impact - Chance fertility in the chemotherapy option decreased by 5% intervals → recording threshold point
Silva et al., 2022 [17]	- Literature - Expertise of specialists	Observational descriptive study: **exploratory interviews** and **online questionnaire**	Not specified	- Ranking from least to most important ○ Treatment outcomes ▪ Adverse effects ▪ Pain ▪ Quality of life ▪ Overall survival ▪ Progression-free survival ○ Impacted aspects of life ▪ Social ▪ Physical ▪ Sexual ▪ Economic ▪ Psychological and emotional
Ballinger et al., 2017 [29]	- Using the profiles of common anthracycline- and taxane-based regimens	A **choice-based conjoint** (CBC) survey	Chemotherapy	- Pairs of treatment choices varying in ○ degree of benefit by risks of recurrence ○ toxicity profile, including peripheral neuropathy and congestive heart failure
Wouters et al., 2013 [36]	- Literature review to determine the themes	Online focus groups and individual interviews with a **Q-sorting task**	Endocrine therapy	- Sixty-three statements and 11 belief items were used in a Q-sorting task
Hollin et al., 2020 [20]	- Literature review - Feedback clinical experts and patients during in-person qualitative interviews	**Discrete choice experiment** (DCE)	Not specified	- Minimum life extension for half of patients compared to current therapy - Average increase in toxicity free days compared to current therapy - Changes in major side effects compared to your current therapy - Treatment requirements - Your monthly out-of-pocket costs - Monthly insurance company costs - Available test to see if the therapy will work for you
Smith et al., 2014 [21]	- Pilot test to check the conjoint analysis design	Online survey containing treatment scenarios for a **conjoint analysis**	Chemotherapy	- Method of administration - Likelihood of benefit (defined as shrinkage of advanced cancer, responding to treatment) - Likelihood of a given side effect
Chou et al., 2020 [19]	- Literature review and expert opinion - Drafted descriptions reviewed by an expert panel followed by a revision of non-medical people	Cross sectional survey: interview with **visual analog scale** (VAS) and **time trade-off** (TTO) to measure health utilities	Not specified	- Health states including ○ Progression-free metastatic breast cancer ○ Responding metastatic breast cancer ○ Progressive metastatic breast cancer ○ Palliative metastatic breast cancer ○ Anemia ○ Arthralgia ○ Diarrhea ○ Fatigue ○ Febrile neutropenia ○ Nausea/vomiting ○ Hand-foot syndrome ○ Stomatitis ○ Thrombocytopenia
Dacosta Dibonaventura et al., 2014 [22]	- Attributes identified from qualitative research with patients and examination of literature	Cross sectional internet-based survey with a part **‘stated preferences and choice task**’	Not specified	- 8 safety attributes ○ Alopecia ○ Motor neuropathy ○ Myalgia/arthralgia ○ Nausea/vomiting ○ Fatigue ○ Neutropenia ○ Mucositis/stomatitis ○ Diarrhea - Effectiveness - Dosing regimen - Quality of life
Liu et al., 2024 [38]	- Selection attributes and levels: literature review and qualitative research, focus group discussion and in-depth interviews - Pilot study with patients to evaluate the questionnaire	Face to face **discrete choice experiment** (DCE)	Central venous access device	- Out of pocket cost - Limitations in activities of daily living - Catheter maintenance frequency - Risk of catheter-related thrombosis - Risk of catheter-related infection - Size of incision
Stamuli et al., 2023 [13]	- Attribute selection: extensive literature review and finalization of attributes and levels via an advisory board - Pilot of the questionnaire: 125 members of the general population	**Discrete choice experiment** (DCE)	Not specified	- Overall survival - Hyperglycemia - Rash - Pain - Functional well-being - Out-of-pocket payment
Simes et al., 2001 [9]	/	Semi-structured interviews with time **trade-off and survival rate questions**	Chemotherapy	- Survival benefit
Galper et al., 2000 [37]	- Design of the survey scenarios by a multidisciplinary team - Pilot tested breast cancer patients and members of the administrative support staff - Comparisons with the literature	Interviews with **hypothetical scenarios**	Axillary lymph node dissection (ALND)	- Hypothetical scenarios for four potential benefits ○ Local control of the disease ○ Survival ○ The impact of information obtained from ALND on treatment recommendations ○ The prognostic information obtained from ALND independent of its effect on treatment recommendations - against the risk of arm dysfunction
Stamuli et al., 2022 [14]	- List of candidate attributes—through a literature review - Discussion in an experts’ advisory board (breast cancer patients, clinicians and former health policy makers) - Think-aloud interviews with two breast cancer patients	**Discrete choice experiment** (DCE)	Not specified	- Progression-free survival - Febrile neutropenia - Pain - Functional well-being - Out-of-pocket payment (per year)
Mansfield et al., 2023 [23]	- Attributes levels: based on available data of approved therapies - Patient advocacy group reviewed and provided input on the study protocol - Qualitative pretested via telephone interviews with eligible participants	**Discrete choice experiment** (DCE)	Targeted therapy	- Progression-free survival - Nausea/vomiting - Diarrhea - Liver function problems - Risk of heart failure - Risk of serious lung damage and infections
Mcquellon et al., 1995 [30]	- Treatment scenarios developed by medical oncologists, nurses, a clinical psychologist, counselors, and patient educators	Interviews with **hypothetical treatment scenarios**	Chemotherapy and endocrine therapy	- Hypothetical clinical scenarios with side effects varying from low to life-threatening
Spaich et al., 2019 [31]	/	Two-part video shown to patients: educational section followed by a **preference elicitation section** and a questionnaire to identify influencing factors	Radiotherapy	- Preference elicitation section focusing on additional acceptance risk of recurrence after either treatment
Reinisch et al., 2021 [24]	- Desk research and qualitative in-depth interviews patients, caregivers, and physicians - Validation of the quantitative phase: pretests with patients to validated and refined wording and comprehensibility of questions/items	Survey with an **adaptive choice- based conjoint** measurement	Not specified	- Therapy goals ○ Gained life time ○ Gained time without disease progression - QoL ○ Emotional balance ○ Participation in social life ○ Physical agility and mobility ○ Flexibility throughout the course of the day/week - Side effects ○ Diarrhea ○ Nausea/vomiting ○ Hair loss ○ Fatigue ○ Dry mucosa ○ Risk of infection
Ngorsuraches et al., 2015 [18]	- Clinical literature of treatments and in-depth interviews with breast cancer patients	**Discrete choice experiment** (DCE)	Not specified	- Progression-free survival - Anemia - Pneumonitis - Monthly treatment cost
Duric et al., 2005 [33]	- Interview piloted and refined with health professionals and breast cancer consumers	Structured, scripted interview using the **trade-off method**	Chemotherapy	- Hypothetical scenarios without chemotherapy on ○ Life expectancies ○ Survival rates
Duric et al., 2005 [32]	- Hypothetical scenarios based on a previous study; based on known potential survival times (5 or 15 years) and rates (60% or 80% at 5 years) without adjuvant endocrine therapy	Semi structured interview with **hypothetical clinical scenarios** with questions on **‘survival time trade-off’** and **‘survival rate trade-off’**	Endocrine therapy	- Hypothetical scenarios without adjuvant endocrine therapy on ○ Potential survival times ○ Potential survival rates
Omori et al., 2019 [26]	- Pilot study to estimate number of valid responses required and determine participants comprehensibility	**Discrete choice experiment** (DCE)	Targeted agents and endocrine therapy	- Progression-free survival - Frequency of stools - Incidence of diarrhea - Duration of diarrhea - Route and frequency of administration of the treatment
Nazari et al., 2021 [27]	- Survey design and development: (1) targeted literature review, (2) interview with clinical experts, (3) pretest (cognitive testing) with a convenient sample of women from the general population, (4) changes double-checked with the medical oncologist clinical consultant, and (5) updated survey tested in an interview-led session with oncologists treating patients with breast cancer	**Discrete choice experiment** (DCE)	Endocrine therapy and targeted therapies	- Progression-free survival - Stomatitis, grade II and III - Neutropenia, grade III and IV - Arthralgia, grade III and IV - Administration mode - Monthly cost
Kuchuk et al., 2013 [11]	- Health states and side effects: based on the literature and medical labeling information for common chemotherapies and breast cancer web forums - Survey pilot tested in five patients	Survey with **standard gamble** questions to obtain preference weights for health states	Chemotherapy	- Grade I/II (mild to moderate) and III/IV (moderate to severe) ○ Diarrhea ○ Hand–foot syndrome ○ Mucositis/stomatitis ○ Nausea ○ Sensory neuropathy ○ Motor neuropathy ○ Fatigue ○ Myalgia - Alopecia
Williams et al., 2021 [12]	- Previous qualitative study: preferences varied widely	**Choice-based conjoint** survey design	Not specified	- Physical side effects - Emotional side effects - Mental side effects - Ability to work - Impact on personal responsibilities - Logistics or convenience - Out-of-pocket events - Impact on activities of daily living - Burden on care partners - Interference with important events - Ability to take part in a clinical trial or use a new medication - Sexual and cosmetic concerns - Fertility
Thill et al., 2016 [34]	- Criteria for the hierarchy model were identified in an earlier qualitative survey	Interviews to determine preferences with **analytic hierarchy process methods**	Neoadjuvant therapy	- Efficacy of the neoadjuvant therapy ○ Destruction of tumor cells ○ Minimization of the risk for tumor recurrence ○ No reduction in life expectancy due to the disease ○ Possibility for breast-preserving operation - Avoidance of side effects of the neoadjuvant therapy ○ Side effects that are stressful for the body ▪ Fever ▪ Diarrhea ▪ Nausea ▪ Fatigue ○ Side effects that cause bodily changes ▪ Loss of hair ▪ Weight gain
Wouters et al., 2013 [28]	- Selection and definition of attributes and level: literature and online focus groups with women treated with endocrine therapy	Online questionnaire or face- to- face interview: **adaptive conjoint analysis** choice task	Endocrine therapy	- Efficacy - Libido decrease - Osteoporosis - Hot flashes - Risk of endometrial cancer - Fluid retention - Joint and muscle pain - Regimen duration
Bullen et al., 2024 [15]	- Attributes and levels: (1) a targeted literature review of qualitative literature concerning the patient experience of metastatic cancer, (2) a targeted literature review of DCEs centered on treatments for metastatic cancer, (3) a thematic analysis of Scottish Medicine’s Consortium Patient and Clinical Engagement statements for mBC treatments, and (4) face-to-face interviews with patients with mBC	**Discrete choice experiment** (DCE)	Not specified	- Fatigue - Nausea - Diarrhea - Additional side effects - Overall survival - Risk of urgent hospital admission
Beusterien et al., 2014 [16]	- Attributes identified from (1) a literature review, (2) a detailed assessment of breast cancer forum discussions, and (3) consultation with clinical experts	Survey with **adaptive conjoint analysis** to elicit preferences	Chemotherapy	- Grade I/II and Grade III/IV ○ Peripheral neuropathy ○ Motor neuropathy ○ Myalgia ○ Nausea ○ Fatigue ○ Hand-foot syndrome ○ Diarrhea - Neutropenia - Alopecia - Administration regimen
Thewes et al., 2005 [35]	- Modifications of previous studies were used to develop the interview	Face-to-face **interview**	Endocrine therapy	- Hypothetical clinical scenarios—questions based on ○ Improvement of life expectancy ○ Improving the probability of survival
Lalla et al., 2014 [25]	- Attributes and levels: literature and collaboration with clinicians	Self-administered **conjoint analysis survey**	Not specified	- Hair loss - Diarrhea - Fatigue - Nausea - Tingling in hands and feet - Pain - Risk of infection - Out-of-pocket costs

**Table 3 cancers-18-00134-t003:** Patient recruitment and patient involvement.

Study	Patient Recruitment		Patient Involvement		
	Recruitment Strategy	Compensation	Patients as Participants Information and Communication		Patients as Co-Researchers Involvement in Study Design
			Before the Study	After the Study	
Tan et al., 2014 [10]	- Identification by the hospital pharmacy - Invitation to participate during consultation or chemotherapy appointments	- Monetary compensation	No	No	No
Srikanthan et al., 2019 [8]	- Not specified	No	No	No	No
Silva et al., 2022 [17]	- Virtual recruitment by the research team in partnerships with patient support from Non-Governmental Organizations	No	No	No	No
Ballinger et al., 2017 [29]	- Online distribution by a non-profit patient support organization (Living Beyond Breast Cancer, LBBC) and an online data collection company (Research Now)	- LBBC participants were not compensated. Research Now participants received points.	No	No	No
Wouters et al., 2013 [36]	- Identification of breast cancer patients by the research team, invitation of the patients via an oncology clinic - Call posted on the website of a patient organization	No	No	No	No
Hollin et al., 2020 [20]	- Via a professional survey panel managed by Survey Healthcare Universal	No	No	No	- Interviews with patients to provide feedback on the attributes
Smith et al., 2014 [21]	- Online recruitment, email via 4 patient advocacy/support organizations	No	- Introductory page and assurance of anonymity	No	- No (pilot test but not clear if this was with patients)
Chou et al., 2020 [19]	- Recruitment of adult breast cancer patients who were followed up at the study cancer center, convenience sampling	No	No	No	No
DaCosta DiBonaventura et al., 2014 [22]	- Cancer-specific online panels → panelists were e-mailed invitations	- Monetary compensation as a donation to a nonprofit charitable organization of their choice	No	No	- Qualitative interviews with patients identifying attributes
Liu et al., 2024 [38]	- Recruitment via oncology departments of three public hospitals in China	No	- Detailed instructions provided on the DCE and the purpose of the survey - In case of difficulties with completing the tasks on their own, patients could complete the survey with the interview’s assistance	No	- Pilot study with patients to evaluate the questionnaire’s difficulty and comprehensibility and identify any inconsistencies
Stamuli et al., 2023 [13]	- Recruitment via a market research company	No	No	No	- Patients as part of the advisory board for the finalization of attributes and levels
Simes et al., 2001 [9]	- Recruitment via the hospital	No	No	No	No
Galper et al., 2000 [37]	- Identification from a database, selection at random from follow-up appointment lists - Introductory letter sent to patients - Women open to participate were contacted by phone	No	- Introductory letter	No	- Breast cancer patients were involved in pilot tests of the scenarios
Stamuli et al., 2022 [14]	- Recruitment via a market research company	- Renumeration	- Introductory page: information on the aim of the survey and lay descriptions of the attributes - Example of a completed choice set and three warm-up choice sets	No	- Patients as part of the experts’ advisory board, discussing candidate attributes - Cognitive debrief, think-aloud interviews with two breast cancer patients
Mansfield et al., 2023 [23]	- Recruitment via an international market research firm via e-mails	No	No	No	- Patient advocacy group reviewed and provided input on the study protocol - Pretesting of the study via telephone interviews with eligible participants (patients)
McQuellon et al., 1995 [30]	- Screening by the attending physicians - Explanation and invitation to participate via phone by the coordinator	No	- Purpose of the study explained to eligible patients via telephone - At the appointment: purpose of the study reviewed and informed consent obtained	No	/
Spaich et al., 2019 [31]	No	No	- Information provided on: the diagnosis, existence of national and international guidelines, and tumor board concept	No	No
Reinisch et al., 2021 [24]	- Via treating physicians, patient organizations, oncology forums, social media, and relevant events	No	- Information prior to the interview - Written consent collected from each respondent	No	- Qualitative in-depth interviews with patients - Pretests of the quantitative online survey
Ngorsuraches et al., 2015 [18]	- Via the hospital	No	- All study details were explained to the patients and informed consent is signed	No	- In-depth interviews with breast cancer patients to verify and refine the attributes
Duric et al., 2005 [33]	- Via five tertiary referral cancer centers in metropolitan Sydney	No	No	No	- Interview piloted and refined with breast cancer consumers
Duric et al., 2005 [32]	- Via 10 UK hospitals	No	No	No	No
Omori et al., 2019 [26]	- Identification from a web panel - Invitation via email	No	No	No	- Pilot study to estimate number of valid responses required and determine participants comprehensibility
Nazari et al., 2021 [27]	- Via Breast Cancer Research Center and Breast Cancer Institute	No	No	No	No
Kuchuk et al., 2013 [11]	- Performed at two Canadian Cancer Centers	No	No	No	- Survey pilot tested in five patients
Williams et al., 2021 [12]	- Email invitation from Patient Advocate Foundation (PAF)	- Monetary compensation as a digital gift card	No	No	No
Thill et al., 2016 [34]	- By personal contact, internet advertisements, information flyers to sport groups and apothecaries, direct enrolment from a database, and contact through an already enrolled person	No	No	No	No
Wouters et al., 2013 [28]	- Via hospitals, community pharmacies, and patient organizations	No	No	No	- Online focus groups with women treated with endocrine therapy for the selection and definition of attributes and level
Bullen et al., 2024 [15]	- Distribution of leaflets at cancer centers and conferences, an online panel, social media engagement with help from breast cancer charities, and a research nurse approaching patients directly during clinic visits	No	No	No	- Interviews and in-person questionnaire piloting sessions with patients informing the final design of the survey
Beusterien et al., 2014 [16]	- Web survey implemented at cancer centers	No	- If assistance for the survey was required, participants could contact the research team member via telephone or email	No	No
Thewes et al., 2005 [35]	- Identification in databases of rural oncology clinics - Information sent to women eligible to participate - Telephoned by a member of the research team	No	- Introductory letter and detailed information sheet - Phone call to answer questions about the study	No	No
Lalla et al., 2014 [25]	- Identification from a consumer panel - Invitation email sent	- Monetary compensation as a gift certificate	No	No	No

## Data Availability

The raw data is available upon request.

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
