# Peer review of "Patient Preferences in Breast Cancer: A Scoping Review"

_cancers, 2025, doi:10.3390/cancers18010134_

Round 1
Reviewer 1 Report
Comments and Suggestions for Authors
A carefully designed review. It provides a comprehensive overview of breast cancer preference studies and offers guidance on the design and conduct of future studies. The aim is to achieve greater generalizability of these types of studies for more effective decision-making in everyday care. Excellent manuscript. I have no critical comments. The topic is important and deserves dissemination.
Author Response
Dear reviewer,
Thank you very much for taking the time to review the manuscript and for the positive report.
Kind regards,
Charlotte Verbeke
Reviewer 2 Report
Comments and Suggestions for Authors
Dear Authors,
First of all, congratulations for your interesting work. I hope that my hints will help you in the next steps of improvement and the final manuscript will be really valuable for the readers. This scoping review systematically maps how breast cancer patient preference studies have been designed, conducted, and reported over three decades. It highlights methodological diversity, inconsistent attribute selection, varying levels of patient involvement, and heterogeneous outcomes. Your review identifies gaps and offers practical recommendations for more standardized, patient-centred preference research.
I've encountered just one concerning weakness: conclusions are descriptive but not actionable. The manuscript states that variability complicates implementation, but does not specify concrete pathways for integration into regulatory, HTA, or clinical decision-making. Can you provide explicit recommendations for regulators, HTA bodies, clinicians, trial designers. Propose a decision framework or checklist for future breast cancer preference studies. Link recommendations to gaps directly identified in your dataset.
Finally, I would like to thank you for the excellent figures and graphs you have prepared for the document, they enhance the value of your work and facilitate the understanding process. would it be possible to create some more graphics to illustrate processes described?
Author Response
Thank you very much for taking the time to review this manuscript. Please find the detailed responses below and the corresponding revisions/corrections highlighted/in track changes in the re-submitted files.
Comment 1: I've encountered just one concerning weakness: conclusions are descriptive but not actionable. The manuscript states that variability complicates implementation, but does not specify concrete pathways for integration into regulatory, HTA, or clinical decision-making. Can you provide explicit recommendations for regulators, HTA bodies, clinicians, trial designers. Propose a decision framework or checklist for future breast cancer preference studies. Link recommendations to gaps directly identified in your dataset.
Response 1: Many thanks for this very valuable and important comment. I tried in part 4.2 to formulate some suggestions based on the identified gaps on how to conduct patient preference studies among breast cancer patients. To respond to your comment, I added some specification for regulators, HTA bodies, clinicians, trial designers. As to the framework I fully agree, this is one of the aims of my PhD project that I will further elaborate on in upcoming work.
Comment 2: would it be possible to create some more graphics to illustrate processes described?
Response 2: Thank you for the comment. May I ask if there are specific elements for which you would like additional graphs or tables? Are there some tables now in the supplementary material that you would suggest adding in the paper?
Reviewer 3 Report
Comments and Suggestions for Authors
This scoping review aims to provide an overview of published studies identifying breast cancer patient preferences. The findings of this study will provide evidence for stakeholders to create a valuable impact for and with patients with breast cancer, either by linking preference studies to clear objectives and implementation strategies or by involving patients throughout the entire process.
The authors should consider the following comments.
- ABSTRACT-BACKGROUND: “focusing on i) the design of the study, ii) preference outcomes including preference heterogeneity, iii) recruitment strategies, and iii) patient involvement,” there are two “iii,” please check.
- ABSTRACT-RESULTS: What are the preferences of patients with breast cancer? This should be reported in this section.
- ABSTRACT-CONCLUSION: The conclusion does not answer the question mentioned in the title.
- “2.2.1. Inclusion and exclusion criteria”: Please provide the inclusion and exclusion criteria separately.
- “3.4. Preference outcomes”: The preference outcomes should be summarized in this section.
No.
Author Response
Thank you very much for taking the time to review this manuscript. Please find the detailed responses below and the corresponding revisions/corrections highlighted/in track changes in the re-submitted files.
Comment 1: ABSTRACT-BACKGROUND: “focusing on i) the design of the study, ii) preference outcomes including preference heterogeneity, iii) recruitment strategies, and iii) patient involvement,” there are two “iii,” please check.
Response 1: Thank you for pointing this out. I adapted this accordingly.
Comment 2: ABSTRACT-RESULTS: What are the preferences of patients with breast cancer? This should be reported in this section.
Response 2: Thank you for the comment, I added this in the results section.
Comment 3: ABSTRACT-CONCLUSION: The conclusion does not answer the question mentioned in the title.
Response 3: Many thanks, I adapted the conclusion accordingly.
Comment 4: “2.2.1. Inclusion and exclusion criteria”: Please provide the inclusion and exclusion criteria separately.
Response 4: Thank you for your comment. I adapted the text accordingly and added more details on the inclusion and exclusion criteria.
Comment 5: “3.4. Preference outcomes”: The preference outcomes should be summarized in this section.
Response 5: Many thanks for pointing this out. Given the different designs and objectives of each individual preference study, I found that it would be inaccurate to generalize and summarize the preference outcomes. In view of your comment, I highlighted some outcomes and further referred to the appendix.